

# Genome-wide identification and expression analysis of NADPH oxidase genes in response to ABA and abiotic stresses, and in fibre formation in *Gossypium*

Gaofeng Zhang[1,*], Caimeng Yue[1,*], Tingting Lu[1,2], Lirong Sun[1] and Fushun Hao[1]

[1] State Key Laboratory of Cotton Biology, Henan Key Laboratory of Plant Stress Biology, School of Life Sciences, Henan University, Kaifeng, Henan, China
[2] Henan University of Animal Husbandry and Economy, Zhengzhou, Henan, China
[*] These authors contributed equally to this work.

Corresponding author
Fushun Hao, haofsh@henu.edu.cn

## ABSTRACT

Plasma membrane NADPH oxidases, also named respiratory burst oxidase homologues (Rbohs), play pivotal roles in many aspects of growth and development, as well as in responses to hormone signalings and various biotic and abiotic stresses. Although Rbohs family members have been identified in several plants, little is known about Rbohs in *Gossypium*. In this report, we characterized 13, 13, 26 and 19 Rbohs in *G. arboretum*, *G. raimondii*, *G. hirsutum* and *G. barbadense*, respectively. These Rbohs were conservative in physical properties, structures of genes and motifs. The expansion and evolution of the Rbohs dominantly depended on segmental duplication, and were under the purifying selection. Transcription analyses showed that *GhRbohs* were expressed in various tissues, and most *GhRbohs* were highly expressed in flowers. Moreover, different *GhRbohs* had very diverse expression patterns in response to ABA, high salinity, osmotic stress and heat stress. Some *GhRbohs* were preferentially and specifically expressed during ovule growth and fiber formation. These results suggest that GhRbohs may serve highly differential roles in mediating ABA signaling, in acclimation to environmental stimuli, and in fiber growth and development. Our findings are valuable for further elucidating the functions and regulation mechanisms of the Rbohs in adaptation to diverse stresses, and in growth and development in *Gossypium*.

## INTRODUCTION

Reactive oxygen species (ROS) like superoxide radical ($O_2^{\bullet-}$) and hydrogen peroxide ($H_2O_2$) are toxic byproducts of plant metabolisms. As signal molecules, they also play pivotal roles in regulating plant growth and development, in responding to hormone signals, and various biotic and abiotic stresses (*Song, Miao & Song, 2014*; *Mhamdi & Breusegem, 2018*; *Qi et al., 2018*).

ROS are generated by both enzymic and non-enzymic reactions. Plasma membrane NADPH oxidases (NOXs), also named respiratory burst oxidase homologues (Rbohs), have been demonstrated to be significant sources of ROS in plants under normal and stress conditions. The Rbohs transfers electrons from cytoplasmic NAD(P)H to $O_2$ to generate $O_2^{\bullet-}$, followed by dismutation of the $O_2^{\bullet-}$ to $H_2O_2$ (*Suzuki et al., 2011*; *Marino et al., 2012*; *Chen & Yang, 2019*). A large body of evidence indicates that Rbohs are vital regulators of many key cellular processes including vegetative and reproductive development, stomatal movement, and responses to hormones and diverse environmental stimuli in plants (*Suzuki et al., 2011*; *Marino et al., 2012*; *Xia et al., 2015*; *Chen & Yang, 2019*; *Sun, Zhao & Hao, 2019*). In *Arabidopsis*, 10 NOX homologs (AtRbohA–AtRbohJ) have been detected. AtRbohB is implicated in modulating seed after-ripening (*Müller et al., 2009*). AtRbohC mainly regulates root hair growth and cellular integrity (*Foreman et al., 2003*; *Macpherson et al., 2008*). Both AtrbohD and AtrbohF play key roles in root formation, stomatal closure, and in adaptation to multiple biotic and abiotic stresses (*Ma et al., 2012*; *Marino et al., 2012*; *Jiao et al., 2013*; *Li et al., 2015*; *Liu et al., 2017*; *Sun et al., 2018*; *Chen & Yang, 2019*; *Wang et al., 2019*). AtRbohE functions in tapetal programmed cell death and pollen development (*Xie et al., 2014*), and AtRbohI acts in drought stress response in seeds and roots (*He et al., 2017*). Both AtRbohH and AtRbohJ exert effects in pollen tube tip growth and polar root hair growth (*Kaya et al., 2014*; *Mangano et al., 2017*). Rbohs have also been identified and investigated in many other plants such as tobacco, potato, tomato, *Medicago* and maize. They regulate plant growth and development, and acclimations to different environmental stresses (*Marino et al., 2012*; *Xia et al., 2015*). In recent years, Rboh gene families have been characterized at genome-wide levels in many plants such as *Arabidopsis*, rice, wheat, *Glycine max*, *Brassica rapa* and multiple fruit trees. The expression patterns of the genes were also studied in tissues and in responding to diverse stresses (*Chang et al., 2016*; *Hu et al., 2018*; *Cheng et al., 2019*; *Li et al., 2019a*; *Li et al., 2019b*; *Liu et al., 2019*; *Navathe et al., 2019*; *Zou, Yang & Zhang, 2019*). Moreover, 2 *RbohBs*, 5 *RbohDs* (3 are *RbohKs* according to our classification) and 4 *RbohFs* in cotton have shown to be differentially expressed after infection by *Verticillium dahliae* (*Li et al., 2019a*; *Li et al., 2019b*). However, knowledge about genomic information and genetic evolution of *Rbohs* in *Gossypium* is lacking, and their expression profiles in response to abiotic stresses in cotton remain to be explored.

Cotton is the most important fiber crop, and greatly contributes to the development of textile industry and national economy. Its growth and development as well as the yield and quality of fibers are significantly affected by various unfavorable environmental factors including drought, salinity and heat stress, and by ROS (*Potikha et al., 1999*; *Allen, 2010*). Therefore, it is great necessary to determine the roles and mechanisms of *Gossypium* Rbohs functioning in acclimation to stresses, and in fiber development.

Here, we conduct a genome-wide and comprehensive survey of the Rboh families in *G. arboretum* (A2), *G. raimondii* (D5), and their derived tetraploid species *G. hirsutum* (AD1) and *G. barbadense* (AD2). The expression patterns of Rboh genes were also analyzed in different tissues, in response to ABA, salinity, osmotic stress and heat stress, and in fiber formation in cotton. The research work will provide valuable information for further investigation of the roles of Rbohs in *Gossypium*.

## MATERIALS AND METHODS

### Identification of *Gossypium* Rboh family members

The amino acid sequences of 10 Arabidopsis Rbohs (AtRbohA-J) were used as queries to search against the genome sequence databases of *G. arboretum* (BGI-CGB v2.0 assembly genome), *G. raimondii* (JGI assembly v2.0 data.), *G. hirsutum* (NAU-NBI v1.1 assembly genome) (http://www.cottongen.org), and *G. barbadense* (https://cottonfgd.org/about/download.html), respectively, applying the BLAST program with default setting (E-value<e$^{-10}$) (*Camacho et al., 2009*). The *Gossypium* Rbohs were confirmed again in NCBI (https://www.ncbi.nlm.nih.gov/Structure/cdd/wrpsb.cgi) and the SMART programme (http://smart.embl-heidelberg.de/) by analyzing the conserved domains. The questionable Rbohs annotations were manually reassessed.

The properties of the *Gossypium* Rbohs were predicted by the online tool ExPaSy (http://web.expasy.org/protparam/). The structures of the *Rbohs* were identified by the GSDS software (http://gsds.cbi.pku.edu.cn). The conserved domains of *Gossypium* Rbohs were determined using the Pfam (http://pfam.xfam.org/) and NCBI web CD-search tool (http://www.ncbi.nlm.nih.gov/Structure/bwrpsb/bwrpsb.cgi). The Adobe Illustrator CC software was used to depict the structures of genes and conserved domains of proteins.

The chromosomal distributions of the *Rbohs* were characterized on the basis of genome annotation described above. The MapInspect software (http://www.mybiosoftware.com/mapinspect-compare-display-linkage-maps.html) was used to visualize the locations of the *Rbohs*.

### Phylogenetic analysis of Rbohs

The genome sequences and gene annotation databases of *Rbohs* were retrieved from the websites for *Arabidopsis thaliana* (http://www.arabidopsis.org), *Theobroma cacao* (http://cocoagendb.cirad.fr), *Ricinus communis* (http://castorbean.jcvi.org), *Populus trichocarpa* (http://www.phytozome.net/poplar), *Glycine max* (https://phytozome.jgi.doe.gov/pz/portal.html#!info?alias=Org_Gmax), *Brachypodium distachyon* (http://plants.ensembl.org/Brachypodium_distachyon/Info/Index), *Oryza sativa* (http://rapdb.dna.affrc.go.jp/), and the four *Gossypium* plants. The amino acid sequences of Rbohs were aligned by the MUSCLE software (https://www.ebi.ac.uk/Tools/msa/muscle/). The phylogenetic trees were made applying the maximum-likelihood (ML) method and the IQ-TREE server, in which the ModelFinder method was used to select the best model. Models JTT+G4 and JTT+R7 were respectively utilized to generate the unrooted phylogenetic tree for the *Gossypium* Rbohs and the multiple species phylogenetic tree of Rbohs. The ultrafast bootstrap approximation was also used (*Minh, Nguyen & Von Haeseler, 2013*; *Nguyen et al., 2015*; *Kalyaanamoorthy et al., 2017*; *Lu et al., 2019*).

### Synteny and *Ka*/*Ks* analysis

The values of nucleotide substitution parameter *Ka* (non-synonymous) and *Ks* (synonymous) for *Gossypium Rbohs* were calculated with the PAML program (http://abacus.gene.ucl.ac.uk/software/paml.html). The homologous genes were searched by the MCScanx software (http://chibba.pgml.uga.edu/mcscan2). The syntenic maps of the *Rbohs*

from *G. arboretum, G. raimondii* and *G. hirsutum* were generated using the circos-0.69 ± 3 software package with default parameters (http://www.circos.ca).

## Expression analysis of *Rbohs* in tissues and in responding to ABA, salinity, osmotic stress or heat stress in cotton

To examine the transcriptional abundances of *Rbohs* in different tissues, samples of roots, stems and leaves were collected from TM-1 cotton plants grown in nutrient soil (rich soil:vermiculite = 2:1, v/v) for 21 d in a growth chamber. Flowers were isolated in the morning at the first day of anthesis. The fibers were obtained from the ovules at about 23 dpa (day post anthesis). To monitor the expression of *Rbohs* in responding to ABA or abiotic stresses, the cotton seedlings were cultivated in liquid 1/2 MS medium for three weeks (*Lu et al., 2017*). The seedlings were then treated with 100 µM ABA, 200 mM NaCl, 10% PEG6000 or high temperature (42 °C) for 3, 6, 12 and 24 h, respectively. The plants without treatment by ABA or a stress were used as controls. The cotton roots for ABA, NaCl or PEG treatment, and leaves for heat stress were collected, immediately frozen in liquid nitrogen and stored at −70 °C. Total RNA extraction, cDNA synthesis and quantitative real-time RT-PCR (qRT-PCR) analysis were performed as described previously (*Lu et al., 2017*; *Zhang et al., 2017*). The specific primers of cotton *Rbohs* for the qRT-PCR experiments were list in Table S1.

## The expression profiles of Rbohs in ovule and fiber development of *G. hirsutum*

The *GhRbohs* transcriptomic data during ovule growth and fiber formation were obtained from the website http://structuralbiology.cau.edu.cn/gossypium/, and the heatmaps were generated using TBtools v0.6672 (*Chen et al., 2018*).

# RESULTS

## Genome-wide survey of Rbohs in four *Gossypium* species

Based on the available database and bioinformatic analysis results, a total of 13, 13, 26 and 19 Rboh members were identified in *G. arboretum*, *G. raimondii*, *G. hirsutum* and *G. barbadense*, respectively (Table 1). The Rbohs in *G. arboretum* and *G. raimondii* were respectively nominated according to their orthologous similarity in amino acid sequences to the 10 Arabidopsis AtRbohs whereas Rbohs in the two tetraploid *Gossypium* genres were named based on their homologous similarity in amino acid sequences to the Rbohs of the two diploid species following the methods of *Mohanta et al. (2015)* and *Zhang et al. (2017)* in other genes. That is, to name a *Gossypium* Rboh gene, we used the first letter of the genus (upper case) and that of the species (lower case), followed by RbohA-J, RbohK or RbohL. The RbohA-J was the same as the name of its ortholog in *Arabidopsis*. RbohK and RbohL were new paralogous members different from Arabidopsis orthologs. If there were several *Gossypium* Rboh orthologs for an *AtRboh* or other Rboh, a hyphen and different numbers (1, 2, 3 or 4) were added to define the different paralogs. For a *Rboh* of *G. hirsutum* or *G. barbadense*, the letter A or D was added at the end of the name to indicate the gene in A or D genome (Table 1).
**Table 1  Rboh family members in *Gossypium*.**

| Gene identifier | Gene name | Size (AA) | Mass (kDa) | pI |
|---|---|---|---|---|
| Cotton_A_29320 | GaRbohB | 884 | 100.8 | 9.19 |
| Cotton_A_13752 | GaRbohD-1 | 933 | 105.3 | 9.04 |
| Cotton_A_33534 | GaRbohD-2 | 916 | 104.1 | 9.13 |
| Cotton_A_18772 | GaRbohD-3 | 872 | 98.6 | 9.12 |
| Cotton_A_05673 | GaRbohE | 1015 | 115.5 | 9.1 |
| Cotton_A_38536 | GaRbohF-1 | 929 | 106 | 9.28 |
| Cotton_A_31123 | GaRbohF-2 | 871 | 99.8 | 9.09 |
| Cotton_A_23171 | GaRbohF-3 | 846 | 95.7 | 9.12 |
| Cotton_A_24667 | GaRbohH | 1273 | 145 | 8.94 |
| Cotton_A_03631 | GaRbohK-1 | 907 | 102.9 | 9.23 |
| Cotton_A_07974 | GaRbohK-2 | 915 | 103.6 | 9.17 |
| Cotton_A_16216 | GaRbohK-3 | 919 | 104 | 8.82 |
| Cotton_A_13344 | GaRbohL | 803 | 91.1 | 9.08 |
| Gorai.007G299500 | GrRbohB | 884 | 100.9 | 9.19 |
| Gorai.009G202500 | GrRbohD-1 | 936 | 105.4 | 9.03 |
| Gorai.002G128200 | GrRbohD-2 | 786 | 89 | 8.94 |
| Gorai.009G273400 | GrRbohD-3 | 930 | 104.7 | 9.13 |
| Gorai.001G017400 | GrRbohE | 943 | 107 | 8.87 |
| Gorai.003G085100 | GrRbohF-1 | 929 | 105.9 | 9.3 |
| Gorai.004G137300 | GrRbohF-2 | 921 | 105.1 | 9.32 |
| Gorai.008G250500 | GrRbohF-3 | 928 | 105.8 | 9.35 |
| Gorai.001G106500 | GrRbohH | 841 | 96 | 9.1 |
| Gorai.001G053300 | GrRbohK-1 | 907 | 102.7 | 9.07 |
| Gorai.008G212100 | GrRbohK-2 | 920 | 104.4 | 9.02 |
| Gorai.003G117900 | GrRbohK-3 | 919 | 103.7 | 8.9 |
| Gorai.008G199100 | GrRbohL | 802 | 90.7 | 9.11 |
| Gh_A11G2426 | GhRbohBA | 884 | 100.7 | 9.22 |
| Gh_D11G2743 | GhRbohBD | 884 | 100.8 | 9.19 |
| Gh_A05G1666 | GhRbohD-1A | 721 | 81.2 | 9.29 |
| Gh_D05G1864 | GhRbohD-1D | 913 | 102.6 | 9.07 |
| Gh_A01G0943 | GhRbohD-2A | 857 | 97 | 8.84 |
| Gh_D01G0990 | GhRbohD-2D | 940 | 107.1 | 8.97 |
| Gh_A05G2211 | GhRbohD-3A | 849 | 94.8 | 9.19 |
| Gh_D05G2471 | GhRbohD-3D | 932 | 105 | 9.06 |
| Gh_A07G0143 | GhRbohEA | 753 | 85.6 | 9.18 |
| Gh_D07G0136 | GhRbohED | 753 | 85.6 | 9.14 |
| Gh_A02G1791 | GhRbohF-1A | 930 | 106 | 9.28 |
| Gh_D03G0688 | GhRbohF-1D | 929 | 105.9 | 9.33 |
| Gh_A08G0982 | GhRbohF-2A | 751 | 86.5 | 9.63 |
| Gh_D08G1257 | GhRbohF-2D | 922 | 105.2 | 9.28 |
| Gh_A12G2669 | GhRbohF-3A | 929 | 105.8 | 9.29 |
| Gh_D12G2750 | GhRbohF-3D | 928 | 105.7 | 9.35 |

**Table 1** (*continued*)

| Gene identifier | Gene name | Size (AA) | Mass (kDa) | pI |
|---|---|---|---|---|
| Gh_A07G0856 | GhRbohHA | 841 | 95.9 | 9.01 |
| Gh_D07G0928 | GhRbohHD | 841 | 95.9 | 9.07 |
| Gh_A07G0398 | GhRbohK-1A | 907 | 102.9 | 9.23 |
| Gh_D07G0463 | GhRbohK-1D | 907 | 102.7 | 9 |
| Gh_A12G1774 | GhRbohK-2A | 918 | 104 | 9.16 |
| Gh_D12G1932 | GhRbohK-2D | 918 | 104.2 | 9.18 |
| Gh_A03G0476 | GhRbohK-3A | 904 | 102.7 | 8.65 |
| Gh_D03G1062 | GhRbohK-3D | 919 | 103.8 | 8.82 |
| Gh_A12G2653 | GhRbohLA | 803 | 91.1 | 9.11 |
| Gh_D12G1807 | GhRbohLD | 802 | 90.8 | 8.99 |
| GOBAR_AA19255 | GbRbohBA | 884 | 100.8 | 9.22 |
| GOBAR_DD12974 | GbRbohBD | 884 | 100.8 | 9.19 |
| GOBAR_AA06210 | GbRbohD-1A | 933 | 105.2 | 8.96 |
| GOBAR_AA29299 | GbRbohD-2A | 942 | 107.3 | 9.13 |
| GOBAR_DD22521 | GbRbohD-2D | 915 | 104.2 | 9.03 |
| GOBAR_AA19212 | GbRbohEA | 951 | 108 | 9.05 |
| GOBAR_DD05976 | GbRbohED | 943 | 107 | 8.87 |
| GOBAR_AA24991 | GbRbohF-1A | 929 | 106 | 9.28 |
| GOBAR_AA08909 | GbRbohF-2A | 878 | 99.6 | 8.91 |
| GOBAR_AA23701 | GbRbohF-3A | 928 | 105.7 | 9.29 |
| GOBAR_AA36532 | GbRbohHA | 872 | 99.9 | 9.15 |
| GOBAR_DD21834 | GbRbohHD | 833 | 95.3 | 9.16 |
| GOBAR_AA28632 | GbRbohK-1A | 880 | 99.7 | 9.41 |
| GOBAR_DD31632 | GbRbohK-1D | 907 | 102.7 | 9.07 |
| GOBAR_AA11222 | GbRbohK-2A | 839 | 95.5 | 9.23 |
| GOBAR_DD20994 | GbRbohK-2D | 985 | 112 | 9.34 |
| GOBAR_AA01568 | GbRbohK-3A | 891 | 100.6 | 8.71 |
| GOBAR_DD02070 | GbRbohK-3D | 914 | 103.3 | 8.83 |
| GOBAR_DD30619 | GbRbohLD | 688 | 78.3 | 9.08 |

The *Gossypium* Rbohs showed conservative physical properties (Table 1). Most of the Rbohs are similar in amino acid (AA) lengths, molecular weights (MWs), and theoretical isoelectric points (pI). There were no significant differences in these parameters among the four species. The GaRbohs, GrRbohs, GhRbohs and GbRbohs had 803-933, 786-943, 721-940, and 833-985 AA, respectively, with the exception of GaRbohE (1015 AA), GaRbohH (1273 AA), and GbRboh-4D (688 AA). The MWs of the Rbohs varied from 78.3 kDa to 145 kDa with an average of 101.3. The pIs of the Rbohs ranged from 8.65 to 9.63 (Table 1).

## Phylogeny and structure of *Gossypium* Rbohs

We constructed an unrooted phylogenetic tree for the 71 *Gossypium* Rbohs, and analyzed their evolutionary relationship between *G. arboreum*, *G. raimondii* and *G. hirsutum*, *G. barbadense*. As shown in Fig. 1A, the Rbohs can be categorized into seven subfamilies (I to VII). Each of the subfamily I, III, and VI contained 6 members. They were respective homologs of AtRbohE, AtRbohB and AtRbohH in the four *Gossypium* species. Both

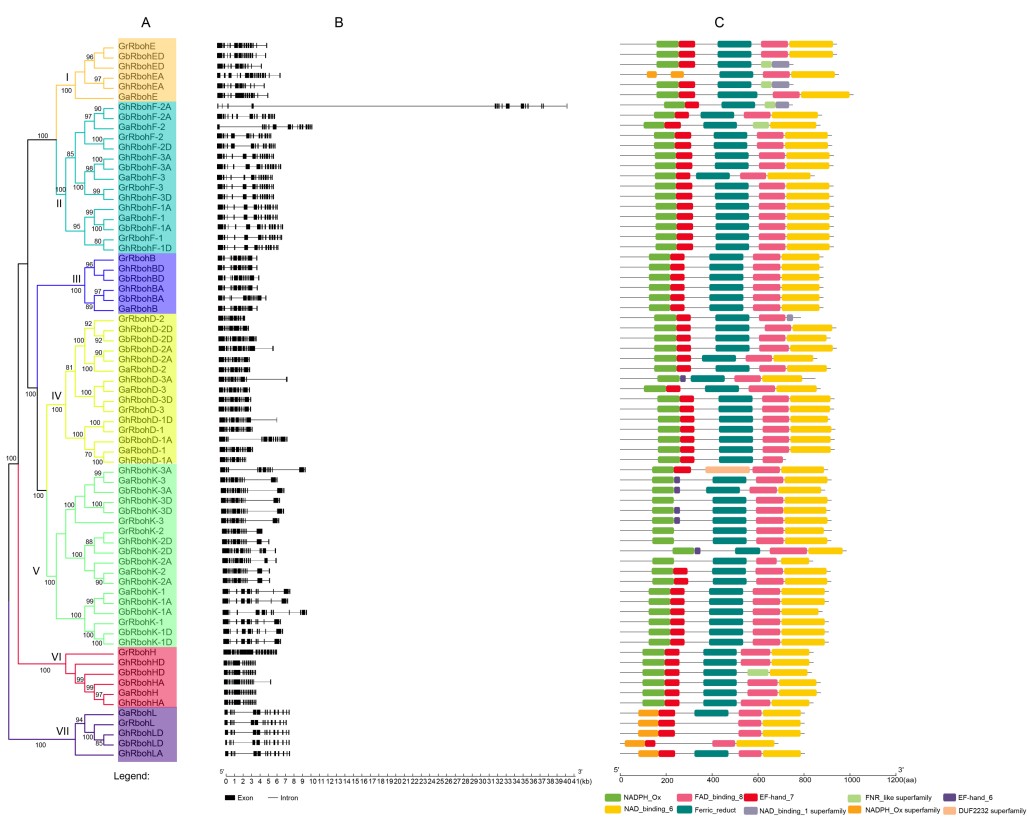

**Figure 1** **Phylogeny, gene arrangements and conserved motifs of Rboh members in four *Gossypium* species.** (A) The phylogenetic tree, (B) Exon/intron structures, the solid black boxes lines are exons and introns, respectively, (C) Profiles of the conserved motifs.

II and IV had 15 members, which respectively belonged to the orthologs of AtRbohF and AtRbohD in *Gossypium*. Subfamily V and VII were composed of 18 and 5 Rbohs, respectively. They were homologous members of AtRbohC and AtRbohJ of *Gossypium*, respectively. Unexpectedly, no orthologs of AtRbohA, AtRbohG and AtRbohI were detected in the tested *Gossipium* plants. We found that all the homologs of an AtRboh in each of the four *Gossypium* species commonly grouped together, reflecting their closer genetic and evolutionary relationship. Moreover, the majority of GaRbohs and GrRbohs had their corresponding homologs in *G. hirsutum* and *G. barbadense*. However, no orthologs of GaRbohD-3 and GrRbohD-3 were detected in *G. barbadense*.

The exon/intron structures of the *Gossypium Rbohs* were analyzed to define the possible evolutionary mechanisms of the genes. It was found that 21, 22 and 15 *Rbohs* individually had 12, 14 and 11 exons. Other Rboh genes harbored 13 (for 6 *Rbohs*), 15 (for 4 *Rbohs*) or 10 (for 3 *Rbohs*) exons, respectively. Generally, the introns of the *Rbohs* were short in length. However, *GhRbohF-2A* possesses a very long intron. Additionally, the *Gossypium Rbohs* in the same subfamily frequently had similar number of exons, and similar exon/intron organizations. For example, all Rboh members of subfamily III contain 12 exons, and each *Rbohs* from subfamily VII carried 14 exons. The exon/intron structures for many

members of one subfamily were alike (Fig. 1B). Regarding the number and organization of exon/intron, about half of *GhRbohs* were very similar to or the same as their orthologs in *G. arboreum* and *G. raimondii*. However, few of *GbRbohs* were very similar to their orthologs in *G. arboreum* and *G. raimondii*, suggesting more distant evolutionary relationship between *GbRbohs* and *GaRbohs* or *GrRbohs*.

To determine the motif compositions, we studied the putative domains in the *Gossypium* Rbohs. Sixty out of the 71 Rbohs had the four characteristic motifs of plant Rboh family (NADPH_Ox, Ferric_reduct, FAD_binding_8 and NAD_binding_6), and most of the Rbohs contained the calcium-binding EF-hand motifs. However, the ferric_reduct motif was absent in GhRbohK-3A, GrRbohL, GhRbohLD and GbRbohLD. Also, the FAD_binding_8 motif did not exist in GhRbohED, GhRbohEA, GhRbohF-2A, GaRbohF-2 and GbRbohHD, and the NAD_binding_6 motif did not exist in GhRbohD-1A (Fig. 1C).

## Chromosomal distributions of *Rbohs*

We analyzed the locations of *Rbohs* in chromosomes from the *Gossypium* plants, and found that the *Rbohs* were unevenly distributed on different chromosomes (Fig. 2). Among these, 10 *GaRbohs*, 13 *GrRbohs*, 22 *GhRbohs* and 19 *GbRbohs* were separately located on 7, 7, 14 and 13 chromosomes, respectively. Three genes were present in each of the D01, D08, At07, Dt07, At′07 and Dt′07 chromosomes. Each of 11 chromosomes (A06, A07, A10, D03, D09, At05, Dt03, Dt05, Dt12, At′12 and Dt′12) harbored 2 genes (Fig. 2). Other chromosomes individually contained one gene. Additionally, the distributions of the *Rbohs* on individual chromosome were irregular. Some genes were situated on the lower end of the chromosome arms, some on the upper end or in the middle region (Fig. 2). Besides, 3 *GaRbohs* and 4 *GhRbohs* were located in scaffolds.

We compared the distributions of *GaRbohs* and *GrRbohs* with their corresponding homologous genes in *G. hirsutum* and *G. barbadense* in chromosomes. Unexpectedly, the majority of the *GhRbohs* and *GbRbohs* did not distribute their corresponding homologous chromosomes in *G. arboreum* and *G. raimondii*. Only the positions of *GhRbohK-3D*, *GhRbohF-1D* and *GbRbohK-3D* in chromosomes were matched with those of *GrRbohK-3*, *GrRbohF-1* and *GrRbohK-3*, respectively (Fig. 2). Our findings suggest that many complex conversion events occur in homoeologous chromosomes harboring *Rbohs* among the *Gossypium* species during the long-term evolution.

## Synteny analysis of *Rbohs*

Gene duplications including tandem and segmental duplications are essential for the expansion of gene family during evolution (*Cannon et al., 2004*). In order to understand the genetic evolutionary relationship among the *Gossypium Rbohs*, we characterized the homologous gene pairs of *Rbohs* from *G. arboreum*, *G. raimondii* and *G. hirsutum*, and analyzed the collinear relationships. A total of 152 homologous pairs were detected among the three species. Twenty-four gene pairs were present between *G. arboretum* and *G. hirsutum*, 45 between *G. raimondii* and *G. hirsutum*, and 25 between *G. arboretum* and *G. raimondii*, 19 between the At-genome and Dt-genome of *G. hirsutum*, and a few within all the *G. arboretum*, *G. raimondii* and *G. hirsutum* species (Fig. 3). These homologous

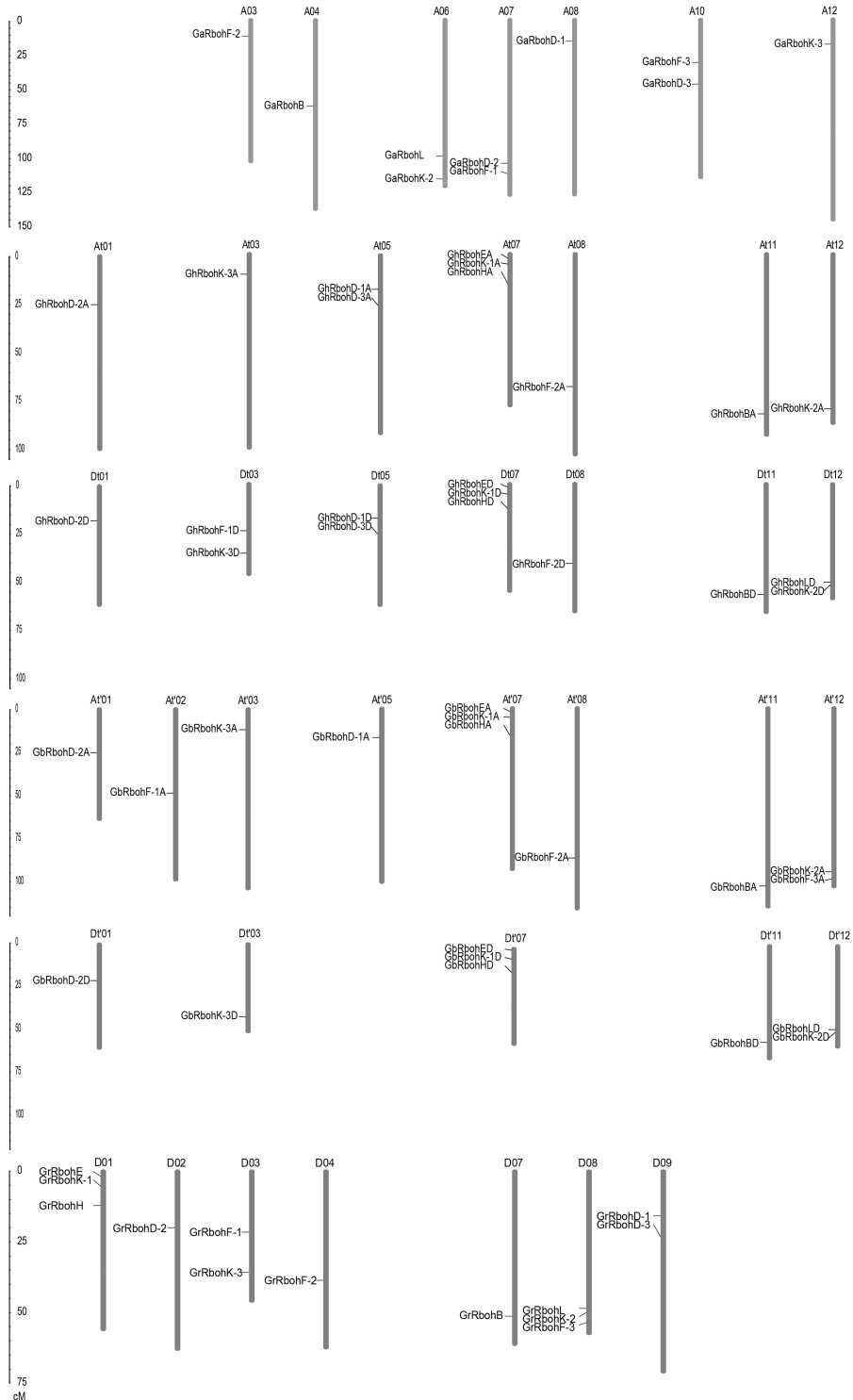

**Figure 2** Distributions of *Rbohs* from *G. arboretum* (*GaRbohs*), *G. raimondii* (*GrRbohs*), *G. hirsutum* (*GhRbohs*) and *G. barbadense* (*GbRbohs*) on chromosomes.

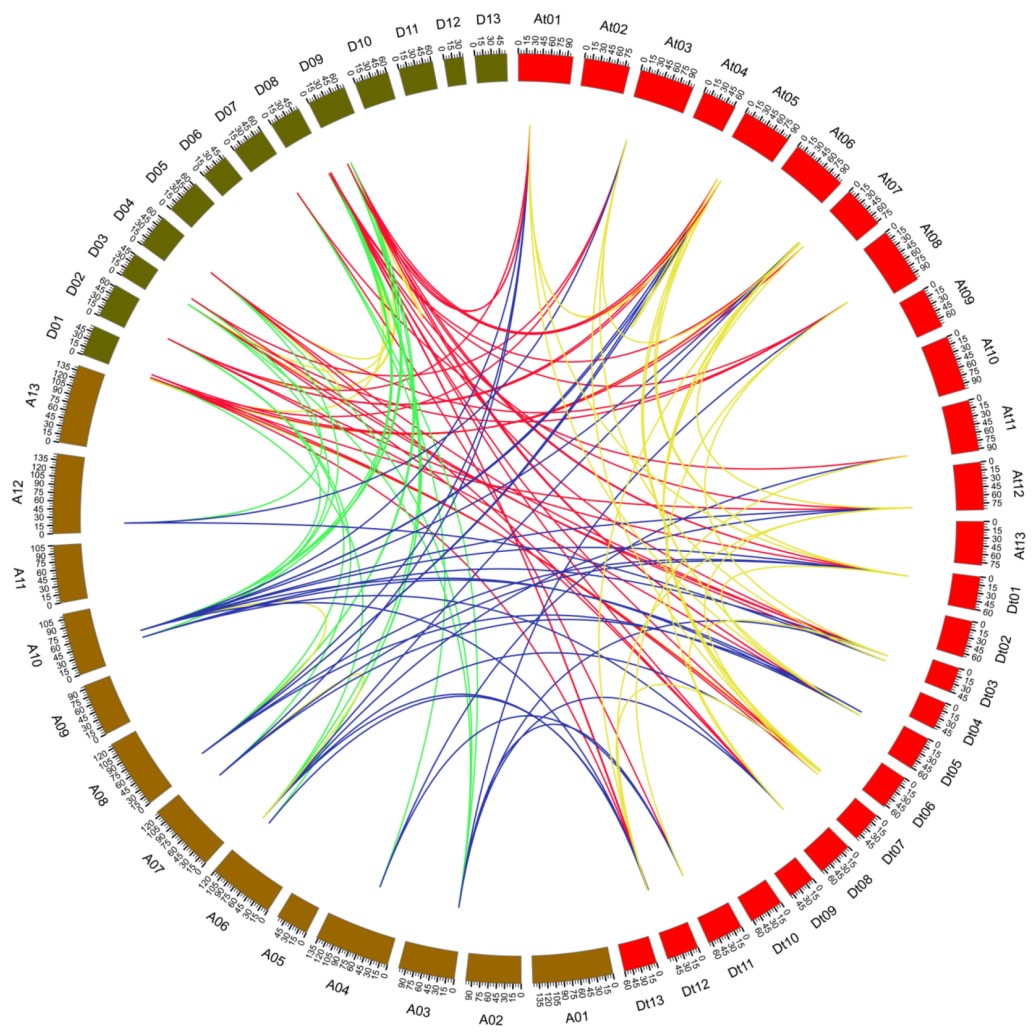

**Figure 3** **Synteny analysis of *Rbohs* of *G. arboreum*, *G. raimondii*, and *G. hirsutum*.** Green lines indicate gene pairs between *G. arboreum* and *G. raimondii*, red lines connected gene pairs between *G. raimondii* and *G. hirsutum*, blue lines show gene pairs between *G. arboreum* and *G. hirsutum*, yellow lines reveal gene pairs within individual species in *G. arboreum*, *G. raimondii*, and *G. hirsutum*.

pairs belonged to 149 collinearity blocks. Most of the blocks owned only one gene pair. However, homologous gene pairs *GrRbohK-1/GhRbohK-1D*, *GrRbohE/GhRbohED* and *GrRbohH/GhRbohHD* between chromosome D01 and Dt07 were in one block, and gene pairs *GrRbohK-1/GhRbohK-1A* and *GrRbohH/GhRbohHA* between chromosome D01 and At07 were in a block. Also, gene pairs *GrRbohK-2/GhRbohK-2D* and *GrRbohL/GhRbohLD* between chromosome D08 and Dt12 belonged to one block, and gene pairs *GhRbohK-1A/GhRbohK-1D* and *GhRbohHA/GhRbohHD* between chromosome At07 and Dt07 belonged to one block. No one gene pair was involved in tandem duplication. These results imply that segmental duplications play predominant roles in the formation of *Gossypium Rbohs* during evolution.

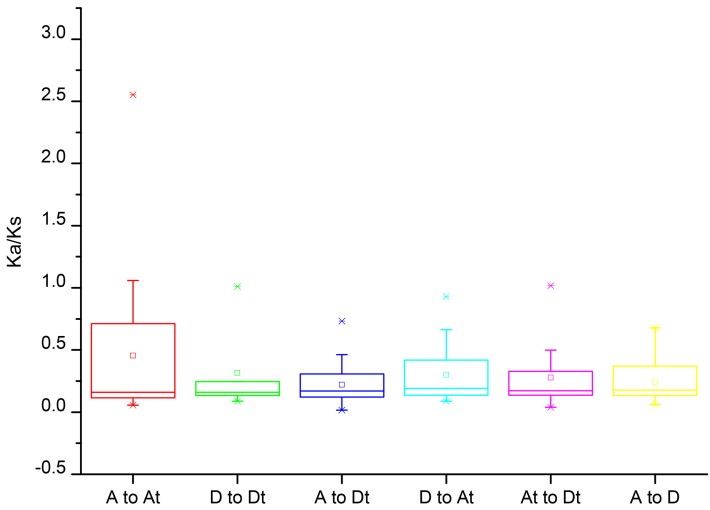

**Figure 4** **The Ka/Ks results of the Rboh homologous genes between A genome, D genome and the subgenomes of *G. hirsutum* (AtDt).**

## Analysis of the *Ka*/*Ks* values of *Rbohs* in *G. arboreum*, *G. raimondii* and *G. hirsutum*

To gain insight into the divergence and selection in duplication of *Rbohs* after polyploidization, the non-synonymous ($K$a), synonymous ($K$s) and $K$a/$K$s values were calculated for the homologous gene pairs among *G. arboreum*, *G. raimondii* and *G. hirsutum*. As shown in Fig. 4, the average *Ka/Ks* values among homologous gene pairs of the *Rbohs* between genomes and/or subgenomes AAt, DDt, ADt, DAt, AtDt and AD were 0.45, 0.31, 0.22, 0.30, 0.28 and 0.24, respectively, and that of all the gene pairs was 0.30, clearly less than 1 (Fig. 4). These data indicate that all the *Rbohs* are mainly under the purifying selection during evolution.

## Phylogenetic relationship of Rbohs in *Gossypium* and other plants

To examine the phylogenetic relationships of Rbohs among the four *Gossypium* species and other plants including *A. thaliana*, *T. cacao*, *R. communis*, *P. trichocarpa*, *G. max*, *B. distachyon*, and *O. sativa*, a phylogenetic tree for Rbohs of these plants was made (Fig. 5). Not surprisingly, most of the *Gossypium* Rbohs were clustered closely in a subfamily. Moreover, the Rbohs from *Gossypium* generally clustered closer with those from eudicots like *A. thaliana* and *G. max* than with those from monocots like *B. distachyon*, and *O. sativa*. The Rbohs from multiple dicotyledon species and those from *B. distachyon* and *O. sativa* were also frequently grouped in a branch (Fig. 5). Notably, the Rbohs from all the *Gossypium* species often clustered closely with those from *T. cacao* (Fig. 5), reflecting the closer evolutionary relationship of *Gossypium* Rbohs with cacao Rbohs.

## Expression profiles of *GhRbohs* in tissues

To better understand the potential roles of *GhRbohs* in different tissues, the expression patterns of the 26 *GhRbohs* genes were studied using qRT-PCR. The results revealed that
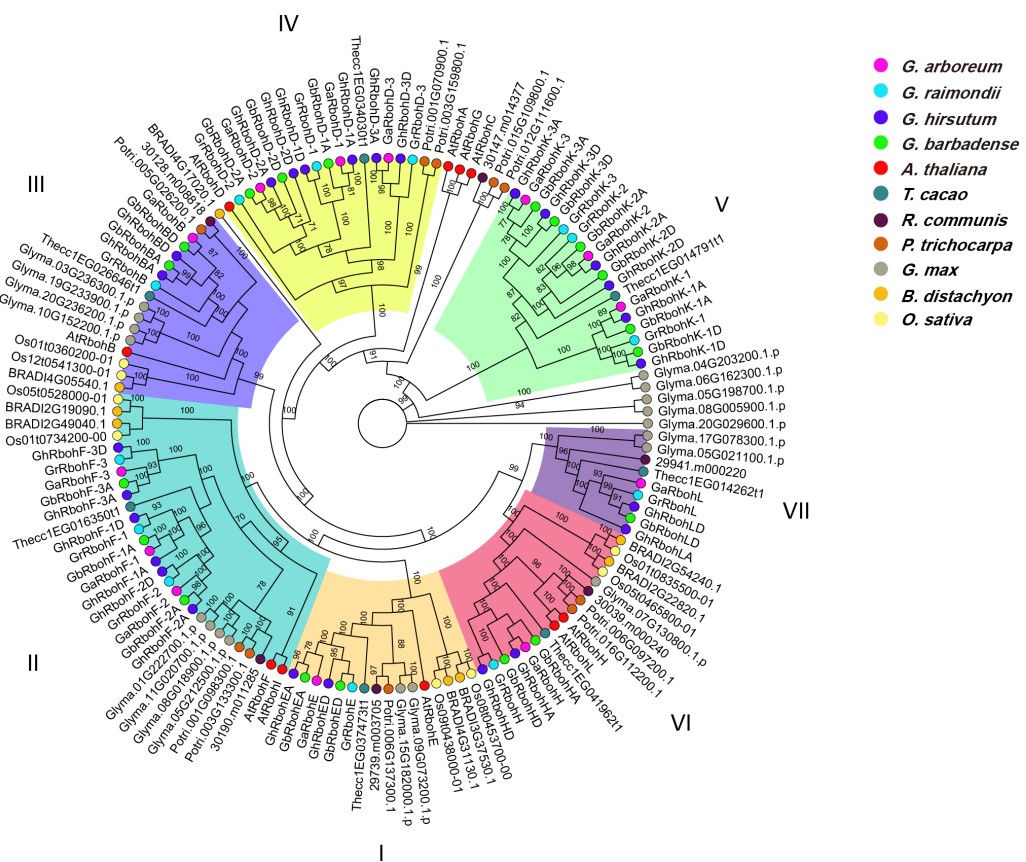

**Figure 5** The phylogenetic tree of Rbohs from four *Gossypium* species, and from *A. thaliana*, *T. cacao*, *R. communis*, *P. trichocarpa*, *G. max*, *B. distachyon*, and *O. sativa*.

all genes were preferentially expressed in flowers (Figs. 6A–6X) except that *GhRbohBD* and *GhRbohK-1D* were highly expressed in roots (Figs. 6Y–6Z). The transcriptional levels of *GhRbohBA*, *GhRbohD-1D* and *GhRbohD-2D* were high in roots whereas those of *GhRbohK-3D* were high in leaves. The transcripts of most genes were not abundant in stems and fibers (Fig. 6, Table S2). These data indicate that the *GhRbohs* might exert key effects in flowers rather than in stems and fibers of cotton.

## Expression patterns of *GhRbohs* in responses to ABA, high salinity, osmotic or heat stress

The transcription changes of the *GhRbohs* were examined after treatments with 100 μM ABA, 200 mM NaCl, 10% PEG6000 or high temperature (42 °C) for different periods of time. In the presence of exogenous ABA, the expression patterns of *GhRbohs* were diverse. The transcriptional levels of 6 genes diminished at early time, but heightened at 12 h and/or 24 h (Figs. 7A–7F). Ten *GhRbohs* showed increased transcription at 3 h or 6 h, and decreased transcription afterwards (Figs. 7G–7P). Expression of 7 genes had decreasing trends in response to ABA (Figs. 7Q–7W). By contrast, the expression of 3 genes

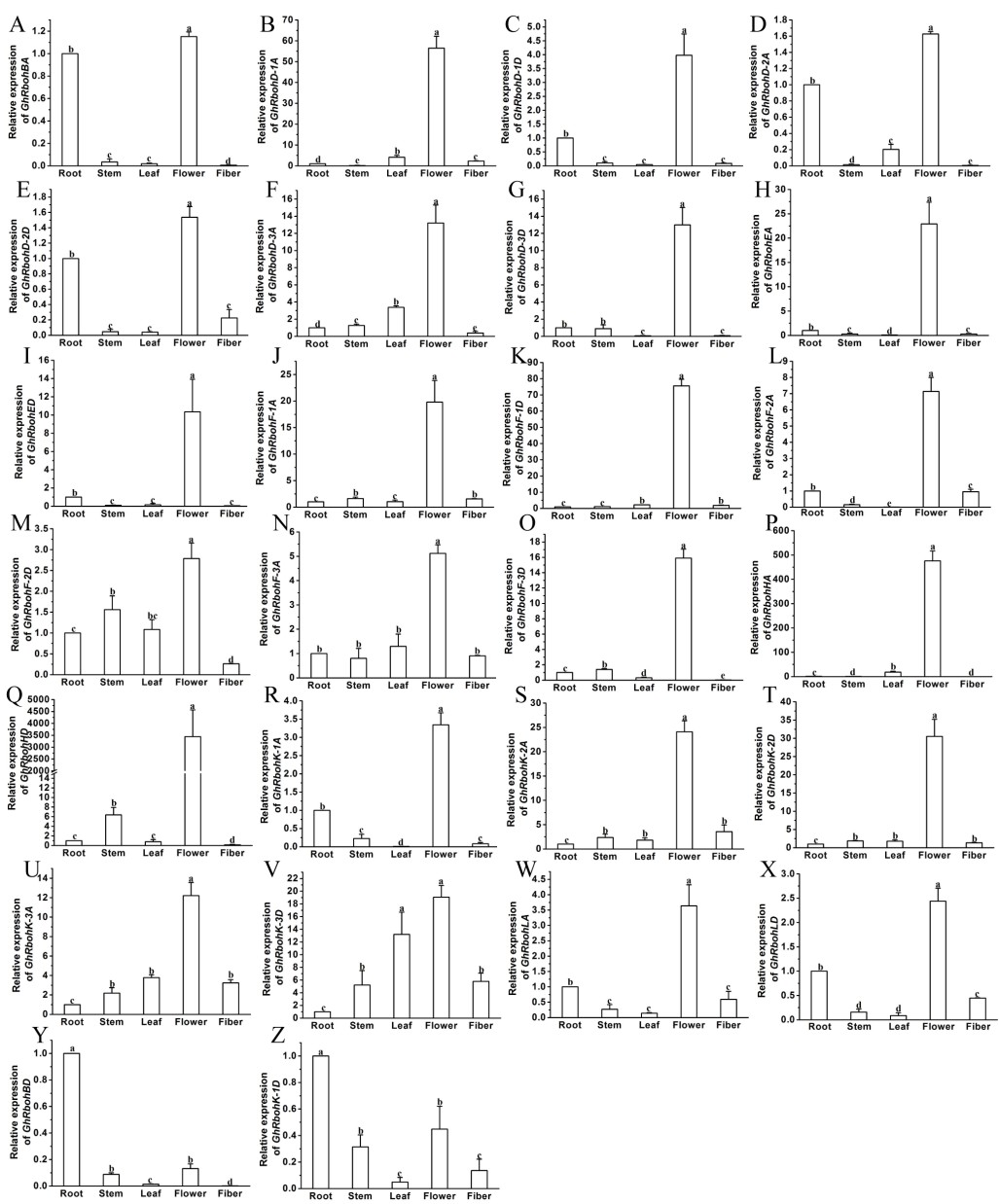

**Figure 6  Expression profiles of 26 *GhRbohs* in distinct tissues of cotton.** The genes are highly expressed in flowers (A–X), and roots (Y–Z). The relative expression of the genes in roots was set as 1. Data are presented as mean ± SE. Different lowercase letters above the error bar indicate significant differences in gene expression levels among diverse tissues by one-way ANOVA and Tukey's HSD test ($n \geq 3$, $P \leq 0.05$).

(*GhRbohK-2A*, *GhRbohF-1A* and *GhRbohLA*) displayed increasing trends in responding to ABA (Figs. 7X–7Z, Table S3).

Under salt stress, the transcriptional levels of 4 *GhRbohs* lessened at 3 h and/or 6 h, but clearly increased at 12 h and 24 h (Figs. 8A–8D). Fourteen *GhRbohs* were upregulated at early time points, and downregulated at 12 h or 24 h (Figs. 8E–8P). The expression of

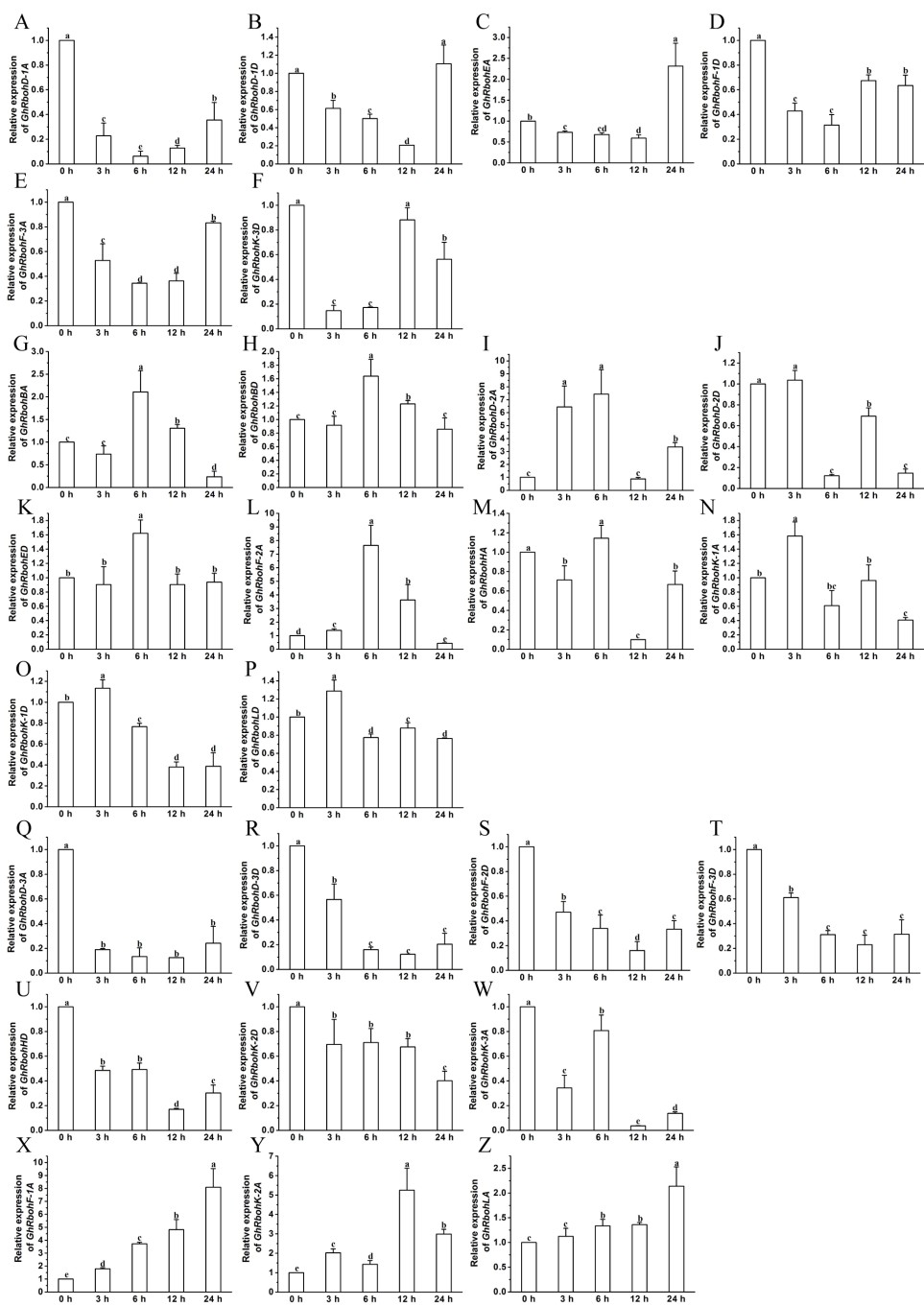

**Figure 7** **Effects of ABA on the expression of *GhRbohs*.** The relative expression of *GhRbohs* was assayed after treatments with 100 µM ABA for the indicated periods of time. The transcript abundances of the genes were rich at 0 h, 12 h and/or 24 h but few at 3 h and 6 h (A–F), were rich in 3 h and/or 6 h, but small at other time (G–P), continually decreased (Q–W), and increased (X–Z). *GhUBQ7* was act as the internal control. The expression value at 0 h was set as 1. Data are presented as mean ± SE. Different lowercase letters above the error bar show significant differences in gene expression levels at different time points by one-way ANOVA and Tukey's HSD test ($n \geq 3$, $P \leq 0.05$).

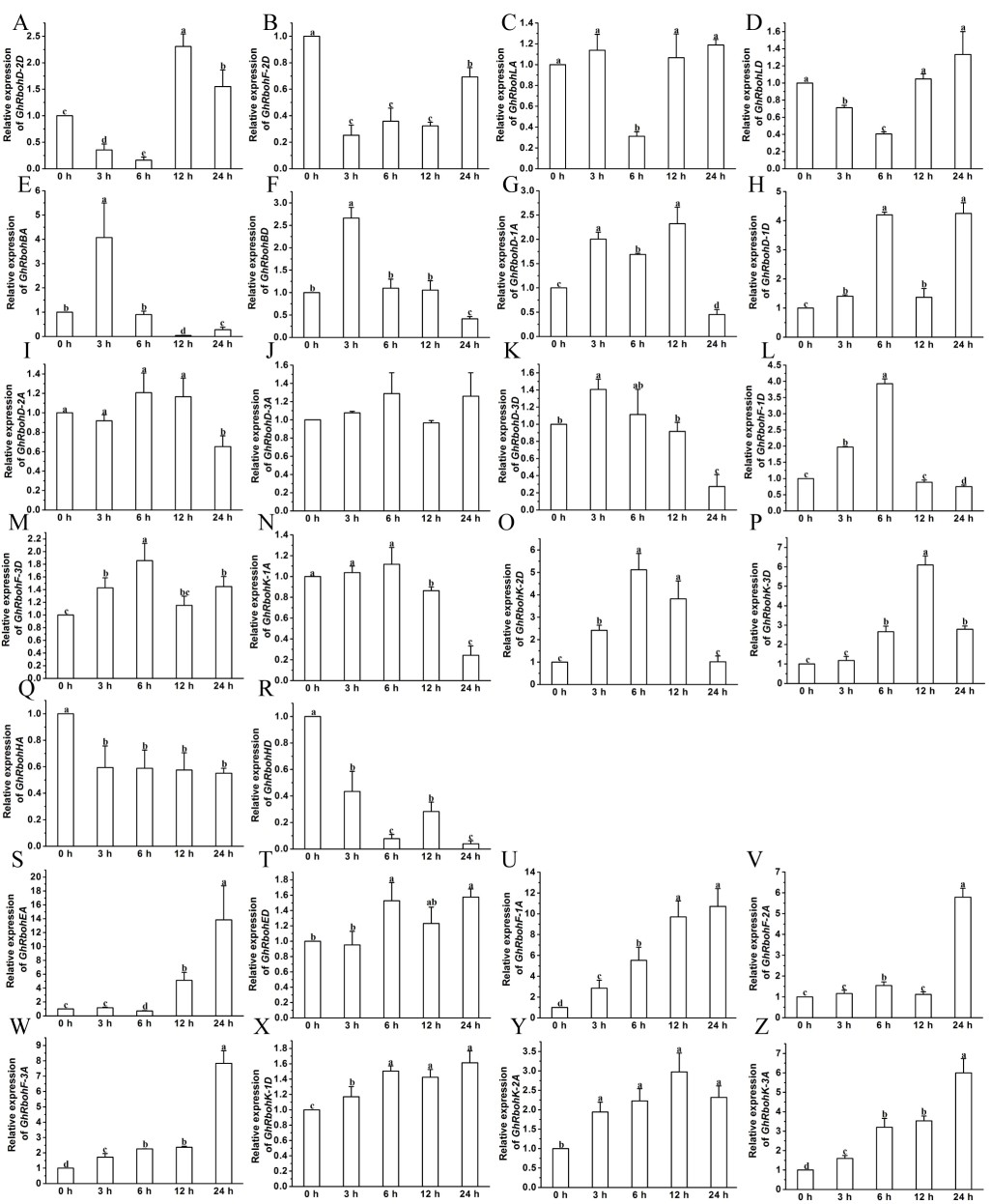

**Figure 8  Effects of 200 mM NaCl on the expression of *GhRbohs*.** The transcription levels of the genes elevated 0 h, 12 h and/or 24 h but diminished at 3 h and 6 h (A–D), increased in 3 h and/or 6 h, but decreased at other time (E–P), continually reduced (Q–R), and enhanced (S–Z). *GhUBQ7* was used as the internal control. The expression value at 0 h was set as 1. Results are presented as mean ± SE. Different lowercase letters above the error bar represent significant differences in transcript abundances at different time points by one-way ANOVA and Tukey's HSD test ($n \geq 3$, $P \leq 0.05$).

*GhRbohHA* and *GhRbohHD* showed diminished trends (Figs. 8Q–8R). In contrast, the abundances of 8 *GhRbohs* exhibited markedly increased trends over treatment time (Figs. 8S–8Z, Table S4).

After treatment of cotton seedlings with PEG, the transcript abundances of 10 *GhRbohs* markedly lowered at 3 h and 6 h, and observably elevated at 12 h and/or 24 h (Figs. 9A–9J). The expression of 7 *GhRbohs* was upregulated at 3 h and/or 6 h, then downregulated over time (Figs. 9K–9Q). The transcriptional levels of 4 *GhRbohs* (*GhRbohK-2A*, *GhRbohD-1D*, *GhRbohD-2A* and *GhRbohED*) showed dropped trends while those of the remaining 5 *GhRbohs* (*GhRbohK-1D*, *GhRbohK-3A*, *GhRbohK-3D*, *GhRbohD-3A* and *GhRbohD-3D*) displayed increased trends over time (Figs. 9R–9U, V-Z) (Table S5).

After exposure of cotton seedlings to high temperature, the expression levels of 5 *GhRbohs* cut down over time till 12 h, but increased at 24 h (Figs. 10A–10E). Eighteen out of twenty-six GhRboh genes were upregulated at 3 h and/or 6 h, but downregulated at 12 h. At 24 h, their transcripts were significantly abundant or few (Figs. 10F–10W). Heat stress caused clear decreases in transcription levels of *GhRbohK-1A* and *GhRbohK-1D* but enhancements in expression levels of *GhRbohF-2A* (Figs. 10X–10Y and 10Z) (Table S6).

Collectively, these results strongly suggest that GhRbohs play differential roles in sensing and transducing ABA signal and in diverse responses to salt stress, osmotic stress and high temperature, highlighting the great importance of NOXs in cotton tolerance to various abiotic stresses.

## Expression patterns of *GhRbohs* in the development of ovules and fibers

To acertain whether GhRbohs play important roles in fiber development, the related transcriptome data were downloaded from the website (http://structuralbiology.cau.edu.cn/gossypium/), and the expression profiles of all the 26 *GhRbohs* during the development of ovules and fibers were investigated. The results showed that *GhRbohK-1A*, *GhRbohK-1D*, *GhRbohK-2A*, *GhRbohK-2D*, *GhRbohK-3D*, *GhRbohD-3A*, *GhRbohD-3D* and *GhRbohF-3D* were highly expressed in the early stages of ovule development, and *GhRbohK-2A*, *GhRbohK-2D*, *GhRbohD-3A*, *GhRbohD-3D* and *GhRbohF-3D* were strongly expressed at some stages of fiber formation till 25 dpa. Some genes like *GhRbohK-1A*, *GhRbohK-1D*, *GhRbohK-3D*, *GhRbohF-1A*, *GhRbohF-1D*, *GhRbohF-3A*, *GhRbohLA* and *GhRbohLD* were expressed at some stages in the development of ovules and/or fibers. Other *GhRbohs* were less expressed or not expressed during growth of ovules and fiber formation (Fig. 11).

## DISCUSSION

In the present study, we identified 13, 13, 26 and 19 Rbohs in *G. arboretum*, *G. raimondii*, *G. hirsutum* and *G. barbadense*, respectively (Table 1). The number of these Rbohs in *Gossypium*, particularly in *G. hirsutum* and *G. barbadense* was larger than that in other plants such as *Arabidopsis* (10), rice (9), maize (14), *Brachypodium distachyon* (9), *Populus trichocarpa* (10), wheat (19), *Brassica rapa* (14) and soybean (17) (*Chang et al., 2016*; *Hu et al., 2018*; *Li et al., 2019a*; *Li et al., 2019b*; *Liu et al., 2019*; *Navathe et al., 2019*), implying that more sophisticated ROS signal transduction mechanisms may exist in *Gossypium*. Compared with other crops as mentioned above, *Gossypium* plants are more tolerant to environmental stresses like drought, high salt, heat stress, and so forth. It is possible that

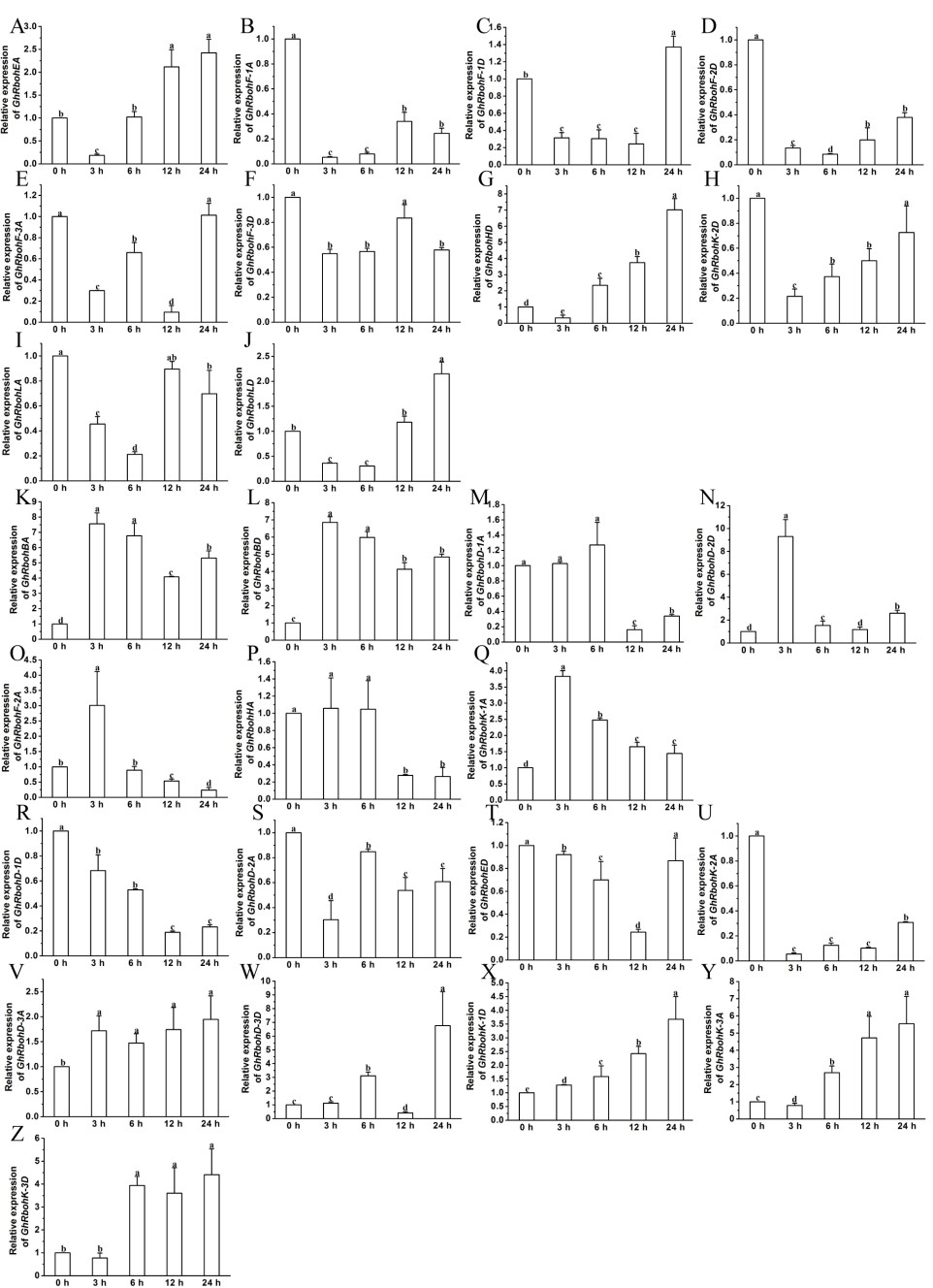

**Figure 9 Effects of 10% PEG on the expression of *GhRbohs*.** The expression levels of the *GhRbohs* increased at 0 h, 12 h and/or 24 h but dropped at 3 h and 6 h (A–J), heightened in 3 h and/or 6 h, but cut down at other time (K–Q), continually decreased (R–U), and increased (V–Z). *GhUBQ7* was applied as the internal control. The expression value at 0 h was set as 1. Data are presented as mean ± SE. Different lowercase letters above the error bar indicate significant differences in gene transcription levels at different time points by one-way ANOVA and Tukey's HSD test ($n \geq 3$, $P \leq 0.05$).

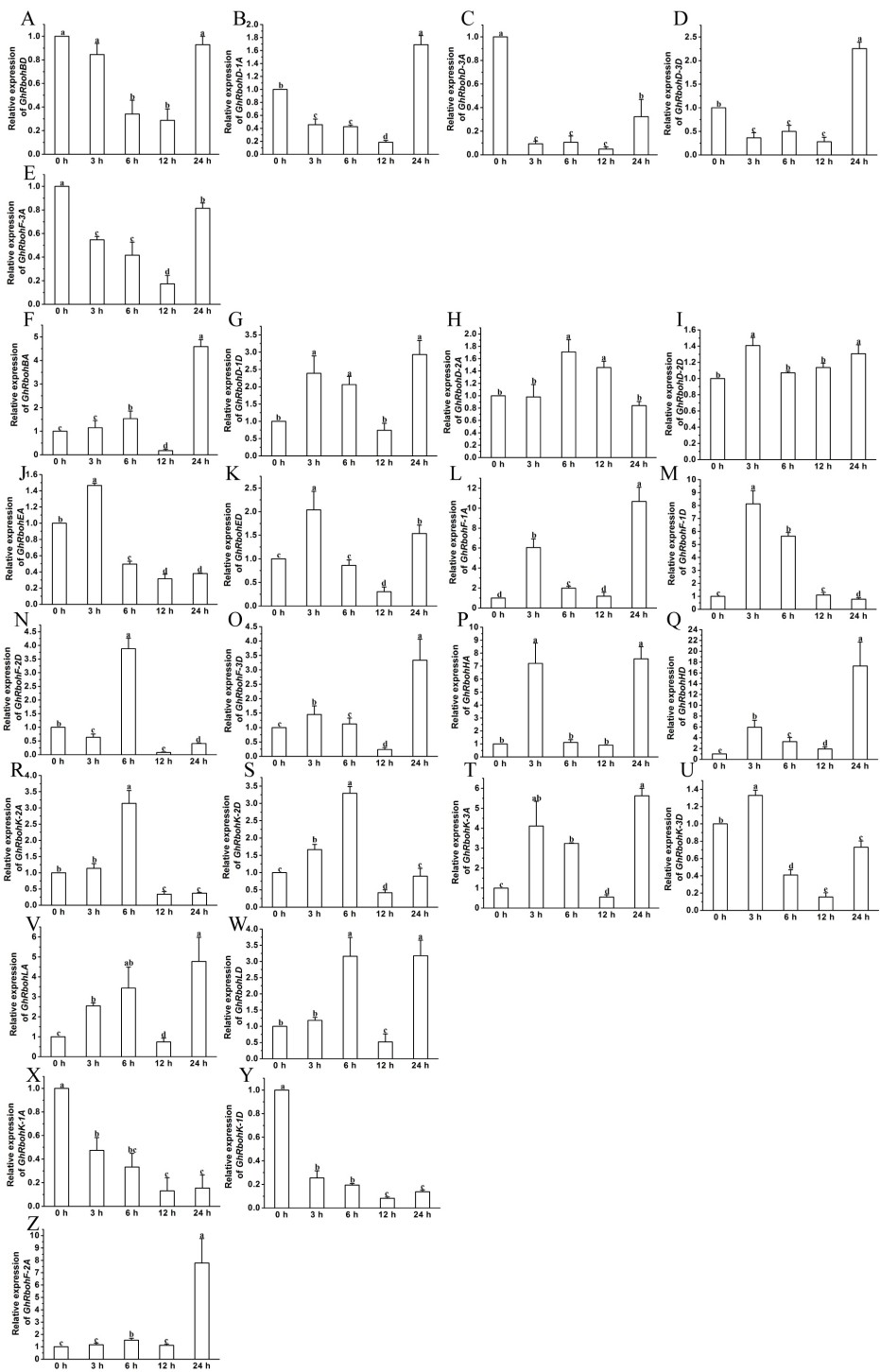

**Figure 10** **The expression profiles of *GhRbohs* under high temperature stress.** The expression of the *GhRbohs* upregulated at 0 h, 12 h and/or 24 h but downregulated at 3 h and 6 h (A–E), increased at 3 h and/or 6 h, but reduced at other time (F–W), continually lowered (X–Y), and elevated (Z). *GhUBQ7* acts as the internal control. The relative expression value at 0 h was set as 1. Data are presented as mean ± SE. Different lowercase letters above the error bar show significant differences in gene expression levels at different time points by one-way ANOVA and Tukey's HSD test ($n \geq 3$, $P \leq 0.05$).

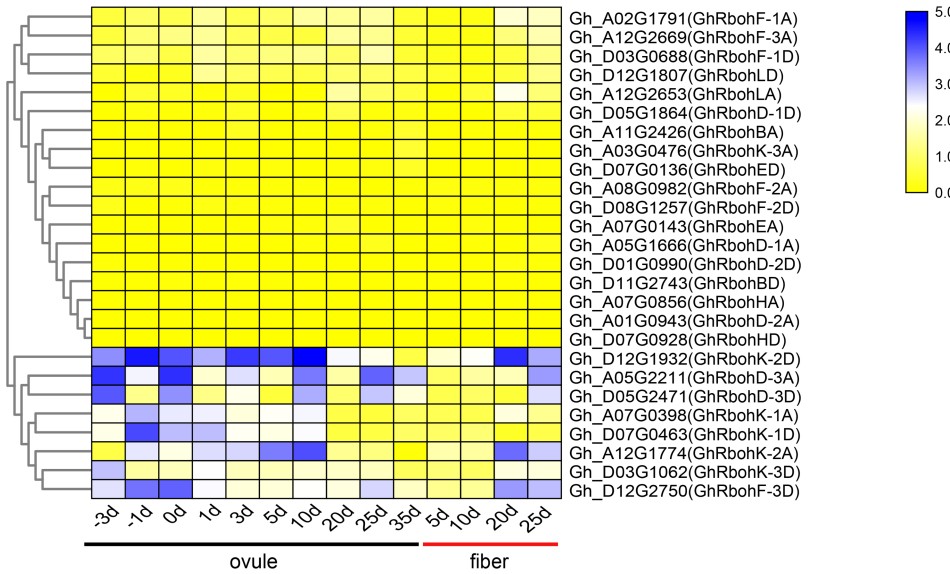

**Figure 11  Changes in expression of *GhRbohs* during development of ovules and fibers in cotton.**  The heat maps in yellow/white/blue (from low to high) indicate the expression levels of various *GhRbohs* at different stages of fiber initiation and elongation.

more Rboh members and complex ROS signals are required for the high tolerance of cotton to stresses.

We observed that all the *GaRbohs* and *GrRbohs* had their corresponding homologues in *G. hirsutum*, and *vice versa*, highlighting the great importance of the *GaRbohs* and *GrRbohs* for survival of *G. hirsutum* after the polyploidization event. In contrast, *GaRbohD-3* and *GrRbohD-3* had no orthologs in *G. barbadense*, indicating that the two genes might not essential for growth and development of *G. barbadense* plants under relaxed selection pressures, and lost during evolution.

The homologous members of each *AtRboh* in the four *Gossypium* species always clustered together, and the exon/intron arrangements of *GaRbohs* and *GrRbohs* were very similar to those of the *GhRbohs* and *GbRbohs* within one subfamily. These suggest that the *Gossypium* Rbohs are originated from a common ancestor. The number of exons for *Gossypium Rbohs* was 10–15, similar to that in *Rbohs* in *Arabidopsis*, rice, maize, wheat, canola and soybean (*Chang et al., 2016*; *Hu et al., 2018*; *Li et al., 2019a*; *Li et al., 2019b*; *Liu et al., 2019*; *Navathe et al., 2019*). Additionally, the majority of the *Gossypium* Rbohs had the conserved EF hand domains, and NADPH_Ox, Ferric_reduct, FAD_binding_8 and NAD_binding_6 motifs of higher plants (Fig. 2). Our findings imply that Rbohs in plants are quite conservative during evolution.

We noticed that each of AtRbohD and AtRbohF had more homologues in the four *Gossypium* species. This coincided with the complex and more diverse roles of AtRbohD and AtRbohF in *Arabidopsis*. Similarly, no orthologs of AtRbohA, AtRbohG, AtRbohI, and AtRbohJ were found in *Gossypium*, in agreement with the small roles of these subunits in *Arabidopsis*. These data indicate that *Gossypium* RbohD and RbohF most likely have more

important roles than other members in growth and development, and in adaptation to stresses.

Identification of gene distributions on chromosomes revealed that most *GaRbohs* and *GrRbohs* did not match with their orthologs of *G. hirsutum* and *G. barbadense* in situated chromosomes and in specific positions, hinting that very complex exchange events of chromosome segments occur in the two tetraploid species after DNA polyploidization.

Synteny analysis showed that most of the *GaRbohs* and *GrRbohs* had many homologous gene pairs in their homologs of the two tetraploid cotton plants, indicating that *Gossypium* Rboh family members amplify mainly through segmental duplication during evolution. To further examine the divergence and selection in duplication of *Rboh* genes, the *Ka/Ks* values were analyzed. The results revealed that the average *Ka/Ks* value for all the homologous gene pairs among *G. arboreum*, *G. raimondii* and *G. hirsutum* was 0.3, much small than 1, suggesting that these genes were majorly under the purifying selection during evolution.

We compared the homologous relationships of Rboh gene family among four cotton species, *A. thaliana*, *T. cacao*, *R. communis*, *P. trichocarpa*, *G. max*, *B. distachyon*, and *O. sativa* using a phylogenetic tree (Fig. 5), and found that the Rbohs from the dicotyledon and monocotyledon commonly gather in one branch, hinting that these Rbohs emerge before the divergence of eudicots and monocots. Moreover, dicot Rbohs always clustered together, and monocot Rbohs also grouped together, indicating that dramatic alterations of Rboh genes occur after the isolation of monocotyledons and dicotyledons. We also observed that *Gossypium* Rbohs clustered closer with Rbohs from cocoa than with those from other plants, reflecting the closer evolutionary relationship of *Gossypium* with cocoa. This means that the majority of homologous Rbohs from *Gossypium* and *cacao* may emerge before the separation of the two genera from one common ancestor.

Transcription analysis showed that most *GhRbohs* were expressed in various tissues including roots, stems, leaves, flowers and fibers. Interestingly, the majority of genes were preferentially expressed in the flower (Fig. 6, Table S2), quite different from the results obtained from *Arabidopsis*, rice, wheat, *Glycine max* and *Brassica rapa* (*Chang et al., 2016*; *Hu et al., 2018*; *Li et al., 2019a*; *Li et al., 2019b*; *Liu et al., 2019*). These suggest that GhRbohs may play key roles in flower development of cotton. The possible reasons for higher expression of *GhRbohs* in flowers might be that: (1) timing flowering was essential for cotton acclimation to adverse stresses, which requires more sensitive and complex ROS signal systems; (2) cotton undergoes long-term human selection for high production of fibers, and for great tolerance to stress conditions, which are closely related to flowering.

The effects of ABA, salinity, osmotic stress or heat stress on the transcript abundances of 26 *GhRbohs* were studied. The results revealed that different *GhRbohs* exhibited very diverse expression patterns in responding to ABA and other abiotic stresses over time (Figs. 7–10, Tables S3–S6), suggesting that each of the *GhRbohs* likely have differential and/or specific functions. These *GhRbohs* may also be spatio-temporally controlled to generate special ROS signals in the presence of ABA or under distinct stresses in cotton. The expression profiles of *GhRbohs* were similar to those of *Arabidopsis AtRbohs* and rice *OsRbohs* in response to drought (PEG), salt and heat stress (*Chang et al., 2016*), and those of soybean *GmRbohs* after treatment with salt or osmotic stress (*Liu et al., 2019*), as well as

those of *Triticum aestivum TaRbohs* in responding to drought and heat stress (*Navathe et al., 2019*), reflecting the great diversity of the expression of *Rbohs* in plants under stresses.

Transcriptome analysis indicated that multiple genes like *GhRbohK-1D*, *GhRbohK-2A*, *GhRbohK-2D*, *GhRbohD-3A* and *GhRbohF-3D* specifically expressed at some stages of ovule growth and fiber development (Fig. 11), implying that these genes may be of very importance for fiber initiation and elongations through producing ROS in cotton.

Stressful conditions like high salinity, drought and high temperature as well as high concentrations of ABA commonly inhibit seedling growth, and promote flowering in plants including cotton (*Takeno, 2016*). The stressors and ABA also stimulate the expression of some *GhRbohs* (Figs. 7–10, Tables S3–S6), and cause the production of ROS in cotton. Therefore, GhRbohs-dependent ROS accumulation induced by the stresses or ABA probably play important roles in the modulation of cotton flowering and subsequent fiber development. ROS derived from NADPH oxidases has shown to act as signal molecules and positively modulate plant responses to diverse stresses (*Suzuki et al., 2011*; *Marino et al., 2012*; *Chen & Yang, 2019*). Environmental stressors generally suppress cotton fiber growth. It seems that ROS generated by the stressors and ABA inhibit fiber elongation. However, cotton has been under long term human selection for both high fiber yields and high tolerance to various stresses. It is conceivable that some stress-affected GhRbohs (most likely GhRbohKs, GhRbohDs and GhRbohFs) may be differentiated in functions and have the ability to positively regulate both fiber development and stress tolerance during the processes of natural evolution and human selection. Some stress-controlled GhRbohs might also exist to negative influence fiber formation in cotton. Yet, the detailed mechanisms need to be thoroughly investigated in the future.

## CONCLUSIONS

A total of 13, 13, 26 and 19 Rbohs were identified in *G. arboretum*, *G. raimondii*, G. *hirsutum* and *G. barbadense*, respectively. These Rbohs were conserved in physical properties, architectures of genes and motifs. The expansion of the Rbohs mainly relied on segmental duplication, and was under the purifying selection. Most GhRbohs were highly expressed in flowers. Different GhRbohs had very diverse expression patterns in responding to ABA, high salinity, osmotic stress and heat stress. Some GhRbohs were preferentially and specifically expressed during the development of ovules and fibers. Our results gave the comprehensive information of the *Gossypium* Rbohs for further research towards understanding the roles of Rbohs in cotton.

### Funding

This work was supported by the Fund of the State Key Laboratory of Cotton Biology, the National Natural Science Foundation of China (31870248), the "111" Project, Science and technology projects in Henan Province (192102110184) and the Foundation of Scientific Research and Innovation team in Henan University of Animal Husbandry and Economy

(2018KYTD18). The funders had no role in study design, data collection and analysis, decision to publish, or preparation of the manuscript.

## Grant Disclosures

The following grant information was disclosed by the authors:

Fund of the State Key Laboratory of Cotton Biology, the National Natural Science Foundation of China: 31870248.

"111" Project, Science and technology projects in Henan Province: 192102110184.

Foundation of scientific research and innovation team in Henan University of Animal Husbandry and Economy: 2018KYTD18.

## Competing Interests

The authors declare there are no competing interests.

## Author Contributions

- Gaofeng Zhang conceived and designed the experiments, performed the experiments, prepared figures and/or tables, and approved the final draft.
- Caimeng Yue performed the experiments, prepared figures and/or tables, and approved the final draft.
- Tingting Lu analyzed the data, prepared figures and/or tables, and approved the final draft.
- Lirong Sun and Fushun Hao conceived and designed the experiments, authored or reviewed drafts of the paper, and approved the final draft.

## Data Availability

The raw measurements are available in the Supplemental Files.

## Supplemental Information

Supplemental information for this article can be found online at http://dx.doi.org/10.7717/peerj.8404#supplemental-information.

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
