# Peer review of "Genome-wide identification and expression analysis of NADPH oxidase genes in response to ABA and abiotic stresses, and in fibre formation in Gossypium"

_PeerJ, doi:10.7717/peerj.8404_

## Round 0.1 · original submission · Major Revisions

Dear Dr. Zhang and Dr. Hao

Thank you for considering the Peer J for your manuscript submission.

I have thoroughly reviewed your manuscript. In my judgment, the topic is highly interesting and state-of-the-art of genes in response to ABA and abiotic stresses; however, it were marked a number of topics by the reviewers that should be addressed to improve the technical quality of your manuscript prior to submission of a revised version of your MS.

When submitting your revised manuscript, please be sure the following points are addressed. Two reviewers raise substantial comments, that need to be addressed adequately.

The first one recommends that should be considerate represent figures 6-10 as tables, for a better results interpretation. A table with similarity values from Blast search (with three best hits) must be added as supplemental material to clarify analysis.
Also, It is widely recommended to perform a statistics analysis of expression profiles/patterns and to describe with more detail the phylogenetic reconstruction.

The second comment relates to The conclusions section; He recommends additional information about the effects of abiotic and ABA treatments. That requests is with the objective to understand the role of Rbohs on growth and development of fiber in Gossypium.

Besides these major concerns, a number of more minor issues considering in the text also need to be addressed. I hope you will be able to address these issues in a next version of this interesting work.

Thank you for your progress, and we look forward to the revised revision.

Reviewer 1 ·

Basic reporting

The text is well write, however need to check some minor changes, particularly in the correct way to write the scientific names.
The references used are adequate.
The results are relevant in understanding the role that Rbohs play during the development of fibers in Gossypium.
Line 168> change “everage” by “average”
Line 174: change “V” by “VI”, in the phylogenetic tree the subfamily VI contained 6 member, while subfamily V contains 18 members.
Line 241: write the scientific names in cursive.
Line 251: change “Cacao” by “cacao”; “Communis” by “communis”; “Trichocarpa” by “trichocarpa”
Line 252: change “Distachyon” by “distachyon”, “Sativa” by “sativa”
Line 255: change “Distachyon” by “distachyon”; “Sativa” by “sativa”
In the Figure 5 change “R. Communis” by “R. communis”
Line 277: use another synonymous of “rose” maybe “increased”
Line 357: change “Cacao” by “cacao”; “Communis” by “communis”; “Trichocarpa” by “trichocarpa”; “Distachyon” by “distachyon”; “Sativa” by “sativa”

Experimental design

The methods are well described.

Validity of the findings

The data shows are clear,

The conclusions are well documented according with the results, however additional information about the effects of abiotic and ABA treatments could help to understand the role of Rbohs on growth and development of fiber in Gossypium.

Additional comments

The present study by Zhang and collaborators “Genome-wide identification and expression analysis of NADPH oxidase genes in response to ABA and abiotic stresses, and in fibre formation in Gossypium”, characterizes NADPH oxidases (Rbohs) in different Gossypium species (G. arboretum, G. raimondii, G. hirsutum and G. barbadense). The results indicated a preferential expression in the floral tissue, which the author suggests a possible role of Rbohs in the fibers formation and abiotic responses.
The document is well write, the authors need to correct the writing of scientific names, the author don`t shows whether the abiotic stress and ABA treatment have an effect on development of fiber in Gossypium only the levels of expression of Rbohs that could help to understand the role of Rbohs.

Reviewer 2 ·

Basic reporting

In this work Gaofeng Zhang et. al. analyze Rboh gene family in four species from Gossypium genus, additionally they evaluate this gene family gene expression in G. hirsutum. The aim and scope of the article is very interesting and relevant. The manuscript is clear and unambiguous, and well writing.

The authors provide enough references and field background; however, the authors could take into account the next references:

Li, P. T., Chen, T. T., Lu, Q. W., Ge, Q., Gong, W. K., Liu, A. Y., ... & Li, S. Q. (2019). Transcriptomic and biochemical analysis of upland cotton (Gossypium hirsutum) and a chromosome segment substitution line from G. hirsutum× G. barbadense in response to Verticillium dahliae infection. BMC plant biology, 19(1), 19.

Wang, F., Chen, Z. H., Liu, X., Shabala, L., Yu, M., Zhou, M., ... & Shabala, S. (2019). The loss of RBOHD function modulates root adaptive responses to combined hypoxia and salinity stress in Arabidopsis. Environmental and experimental botany, 158, 125-135.

Zhang, F., Zhu, G., Du, L., Shang, X., Cheng, C., Yang, B., ... & Guo, W. (2016). Genetic regulation of salt stress tolerance revealed by RNA-Seq in cotton diploid wild species, Gossypium davidsonii. Scientific reports, 6, 20582.

The structure of the article has a standard format. The figures are relevant and help to understand the paper. However, the author should be considerate represent figures 6-10 as tables, for a better results interpretation. The paper presents enough ‘self-contained’ results to represent an appropriate ‘unit of publication’.

Experimental design

Bioinformatics and experimental findings are original and within Aims and Scope of the journal. The identified knowledge gap (knowledge about genomic information and genetic evolution of Rbohs gene families in Gossypium genus) is well filled, following high technical standard. However, the phylogenetic reconstruction should be better described.

Validity of the findings

The majority of the results and findings are well supported. However, there are some issues that must be clarified.

Gene identification was made using similarity, nevertheless in the multiple species phylogeny (Figure 5), this classification did not totally match, for example, F4 genes (Family VII in Figure 1) clusters with AtRbohH (and AtRbohJ), so this suggest that this group might be RbohH or RbohJ-like proteins. F cluster (family II) groups whit AtRbohF and AtRbohI, so this is not clear if family II might be RbohF or RbohF proteins. A table with similarity values from Blast search (with three best hits) must be added as supplemental material to clarify last two points.

Family number legends from Figure 1 must be show in Figure 5.
OTUS legends in Figure 5 must be the same that legends used in Figure 1.
Bootstrap values (>70) in Figures 1 and 5 must be showed.

It is widely recommended to perform a statistics analysis of expression profiles/patterns experiments to determine significant differences between treatments.

---

## Round 0.2 · Minor Revisions

Please, attend to the observation that the reviewer 2 correctly indicates; the recommendation will be important to show an additional highlight for your work; however, you have the last word.

Reviewer 1 ·

Basic reporting

No comment

Experimental design

No comment

Validity of the findings

No comment

Reviewer 2 ·

Basic reporting

In this work Gaofeng Zhang et. al. analyzed Rboh gene family in four species from Gossypium genus, additionally they evaluate this gene family gene expression in G. hirsutum. The aim and scope of the article is very interesting and relevant. The manuscript is clear and unambiguous, and well writing.

The authors provide enough references and field background and include those ones that were recommended.

The structure of this document version has a standard format. The figures are relevant and help to understand the paper and added information that was required as supplementary materials.

The paper presents enough ‘self-contained’ results to represent an appropriate ‘unit of publication’.

Experimental design

Bioinformatics and experimental findings are original and within Aims and Scope of the journal. The identified knowledge gap (knowledge about genomic information and genetic evolution of Rbohs gene families in Gossypium genus) is well filled, following high technical standard.

In this version, the authors describe the methodology in a good manner as it was required.

It is recommended to perform an anova statistical analysis.

Validity of the findings

The majority of the results and findings are well supported. The authors made the corrections that were required. However, the new information has allowed us to observe some issues that must be addressed before the document is accepted.

1. Phylogenetic analysis (Figure 5) and Blast results (Table S7, lines 43-45-Cotton_A_13344, and lines 129-132, Gorai.008G199100) showed that group VII (Gossypium Rbohs J) formed a new paralogous group since AtRbohH and AtRbohJ clustered together with Gossypium Rbohs group VI (Gossypium Rbohs H); group VI and VII are well supported by a high bootstrap values, suggesting that are reliable groups. Furthermore, group VII proteins domains analysis (Figure 1C) sustain this idea. In this case, the sequence alignment length of the Rbohs with AtRbohJ and the Maximum-likelihood values (that must be showed) do not validate the result presented since this values can be explained due to evolutive distance of all AtRboh proteins, therefore, tree topology must be taken in count and Gossypium Rbohs J must be renamed (as author decided) as a new kind of Rbohs proteins.

2. In the same way tree topology showed that Gossypium Rbohs C (group V) is also a new paralogous group and must be renamed as a new kind of Rbohs proteins.

3. It is widely recommended performing an ANOVA statistical analysis of expression profiles/patterns experiments to determine significant differences between treatments since with Student’s t test it is not possible to analyze between different tissue and time.

Minor issue:
• In lines 341-342 is mentioned that GaRbohJ, GrRbohF-1, GrRbohF-3 and GrRbohD-1 had no orthologs in G. barbadense, however list of proteins appear GbRbohJD, GbRbohF-1A, GbRbohF-3A and GbRbohD-1A, this proteins and tree topology shows that are orthologs.

---

## Round 0.3 · accepted · Accept

I inform you that the MS has completed the peer review successfully. All questions and recommendations were satisfied. I suggest it can be accepted.